# Functional Interdependence in Coupled Dissipative Structures: Physical Foundations of Biological Coordination

**DOI:** 10.3390/e23050614

**Published:** 2021-05-15

**Authors:** Benjamin De Bari, Alexandra Paxton, Dilip K. Kondepudi, Bruce A. Kay, James A. Dixon

**Affiliations:** 1Center for the Ecological Study of Perception and Action, University of Connecticut, Storrs, CT 06269, USA; alexandra.paxton@uconn.edu (A.P.); dilip@wfu.edu (D.K.K.); brucekay42@gmail.com (B.A.K.); james.dixon@uconn.edu (J.A.D.); 2Department of Psychological Sciences, University of Connecticut, Storrs, NC 06269, USA; 3Department of Chemistry, Wake Forest University, Winston-Salem, NC 27109, USA

**Keywords:** self-organization, dissipative structures, collective behavior, coordination, coordination dynamics, thermodynamics, maximum entropy production

## Abstract

Coordination within and between organisms is one of the most complex abilities of living systems, requiring the concerted regulation of many physiological constituents, and this complexity can be particularly difficult to explain by appealing to physics. A valuable framework for understanding biological coordination is the *coordinative structure*, a self-organized assembly of physiological elements that collectively performs a specific function. Coordinative structures are characterized by three properties: (1) multiple coupled components, (2) soft-assembly, and (3) functional organization. Coordinative structures have been hypothesized to be specific instantiations of *dissipative structures,* non-equilibrium, self-organized, physical systems exhibiting complex pattern formation in structure and behaviors. We pursued this hypothesis by testing for these three properties of coordinative structures in an electrically-driven dissipative structure. Our system demonstrates dynamic reorganization in response to functional perturbation, a behavior of coordinative structures called *reciprocal compensation*. Reciprocal compensation is corroborated by a dynamical systems model of the underlying physics. This coordinated activity of the system appears to derive from the system’s intrinsic end-directed behavior to maximize the rate of entropy production. The paper includes three primary components: (1) empirical data on emergent coordinated phenomena in a physical system, (2) computational simulations of this physical system, and (3) theoretical evaluation of the empirical and simulated results in the context of physics and the life sciences. This study reveals similarities between an electrically-driven dissipative structure that exhibits end-directed behavior and the goal-oriented behaviors of more complex living systems.

## 1. Introduction

The coordination of action demonstrated by organisms requires, for even the most modest movements, the control of immense numbers of physiological degrees of freedom [1,2]. The scope of this achievement is magnified when one recognizes that this coordination happens in real time within a changing environment. Biology is nevertheless quite adept at performing under these conditions—organisms generally coordinate their activities with great facility. This coordination extends between organisms as well, for example, in collective foraging of slime-molds [3], collective decision-making in bees [4], and human interpersonal coordination [5]. Human communication requires coordination of verbal and non-verbal activities [6] (e.g., gaze, gesture, and posture), and these embodied aspects of coordination have rich social and cognitive consequences, such as rapport building [7], intergroup bonding [8], and joint action [9]. A valuable concept for understanding intra- and inter-personal coordination is the *coordinative structure*, self-organized ensembles of physiological constituents. Self-organization is a well-studied phenomenon in the physical sciences, wherein structures spontaneously emerge in open systems held out of equilibrium. Herein, we report observations of functionally coordinated behavior in a non-living self-organizing system derived from intrinsic end-directedness to maximize the rate of entropy production.

### 1.1. Coordinative Structures

Researchers refer to coordinated assemblies of physiological constituents as *synergies* or *coordinative structures* to emphasize their stability and flexibility [5,10,11]. At the macroscopic level, coordinative structures have been shown to have three properties that support effective action: (1) bi-directionally coupled constituents: this property requires that the physiological constituents (or individual organisms) within a coordinative structure are constrained such that their dynamics are mutually dependent. (2) Soft assembly or dynamic stability: this property holds that the coordinative structure be flexible for changing contexts [10,12]. (3) Functionally specific organization: this property requires that the constituent degrees of freedom be organized relative to some functional end (e.g., the fingers of the hand being organized for the purpose of grasping an object) [11,13].

These properties are dramatically illustrated in a phenomenon known as *reciprocal compensation* [5,10], wherein a physiological ensemble adapts its behavior to compensate for perturbations. Reciprocal compensation was originally observed in speech by Kelso et al. [10]. A participant was asked to repeatedly produce an utterance and was intermittently (and unpredictably) perturbed by a mechanical force applied to the jaw. The perturbation prevented the lower lip from moving to its usual position (i.e., raising to ensure lip closure). Critically, however, the movement of the upper lip immediately compensated for the perturbation to the lower lip, thereby completing the performance of the intended utterance. Reciprocal compensation, then, results when mutually constrained constituents (property 1) spontaneously reorganize their activity to account for changing contexts (property 2) and thus preserve functionality (property 3). This phenomenon has been demonstrated in the production of grip forces among an individual’s fingers [14], an individual’s oscillating limbs [15], and interpersonal activity [5,15]. Observation of reciprocal compensation in a system thus serves as compelling evidence that the system functions as a coordinative structure.

### 1.2. Self-Organization

Self-organization in terms of oscillating chemical reactions (chemical clocks), as well as static and dynamic patterns, has been studied for some time [16,17,18]. Recently, our group has investigated the life-like *behaviors* of non-living dissipative structures, including a surprising end-directedness akin to that observed in living organisms [19,20,21]. In this study, we continue this approach, demonstrating that a non-living self-organizing system can exhibit the same dynamics as biological coordinative structures.

The properties of coordinative structures hold across phyla [22] and even appear to hold when multiple organisms are involved [5,23]. The ubiquitous nature of coordination suggests the need for a general framework, one that is not bound to specific aspects of anatomy and physiology. To this end, many theorists have suggested that coordinative structures emerge through self-organization, wherein the mutual activity of many semi-autonomous elements, for example, motor units, nerves, or people, drives emergent macroscopic order [20,24,25,26]. In this view, coordinated action relies on self-organization in the spontaneous generation and manipulation of constraints on the physiological degrees of freedom [1,2,27]. These constraints produce linked assemblies of physiological constituents (or individual organisms) [5,11,13,15,23,28].

Self-organization, as such, is understood to be a non-equilibrium process, wherein nonlinear relations between thermodynamic forces and flows drive the emergence of constraints [20,26]. The structures formed by such processes are called *dissipative structures*; canonical examples include Rayleigh-Benard convection cells and autocatalytic oscillating chemical reactions [17,20,26]. Living systems are dissipative structures, sustained by a thermodynamic flux of metabolic materials that drives self-organization of morphologies and behaviors [20,29,30]. Importantly for current considerations, coordinative structures have been hypothesized to be a type of dissipative structure [28], with some of their capabilities owing to the underlying thermodynamic contingencies.

### 1.3. A Simple Self-Organizing System: E-SOFI

The system employed in the current study is a variation of a self-organizing electrical system reported in previous works [19,21,31,32]. The system consists of metal beads in shallow oil subject to a high electrical voltage. A source electrode is positioned 5 cm above the dish, separated by an air gap, and a circular grounding electrode is fixed in the dish (Figure 1). Charges from the source electrode collect on the beads, which become dipoles and are attracted to the grounding electrode. This drives the formation of strings of beads that we refer to as *trees* (due to their morphology; Figure 1).

These trees maintain their configuration and contact with the grounding electrode. They tend to oscillate by pivoting on the base bead that is in contact with the grounding electrode. The oscillation is driven by the cyclic accumulation of charges on the oil surface, and the depletion (i.e., conduction to ground) of charges by the trees [31]. The formation of the trees and their behavior is such that the rate of entropy production (REP) Σ increases [19,21,31]. REP in this system is calculated as:(1)$$\Sigma =\frac{V*I\left(x,t\right)}{T}\;$$
where *V* is the voltage (held constant by the power supply), *I* is the current that depends on the location of the tree *x* and time *t,* and *T* is the temperature. Voltage *V* and temperature *T* are constant throughout trials, meaning that the REP is a scalar multiple of the measured current values *I* (although the applied voltage is around 26 kV, since the current is typically between 1–2 μA, ohmic heating is very low and thus T is effectively constant in this system). The results presented below refer to the current through E-SOFI trees, though the same conclusions will apply to the REP. This system’s behavior is rudimentarily end-directed towards states of maximal REP [19,21,31]. The tree structures depend on the flow of charges to maintain integrity, and so behaviors that increase access to charges (and consequently increasing the REP) are *functional* in that they support the continued existence of the structures. This behavior—moving to ensure access to energetic resources that maintain system stability—is analogous to *foraging* in organisms. We thus have called the system the electrical self-organized foraging implementation (E-SOFI).

Applications of a maximum entropy production principle (for a review see [33]) have been demonstrated to be predictive in contexts of global climate modeling [34,35], fluid flow [36], chemical pattern-formation [37], and even bacterial communities [38,39]. Here, we investigated whether this intrinsic end-directedness supports the coordinated behavior of coupled dissipative structures. Given that the trees share the embedding electrical field and share an intrinsic tendency to optimize the REP, we hypothesized that the joint activity of the trees would similarly maximize the REP.

Davis et al. [40] demonstrated that multiple E-SOFI trees can exhibit functionally coordinated activity, coupled through a shared distribution of charges on the oil surface, and that this coordination was directly related to the maximization of the REP. Two trees were placed in the dish and allowed to settle into steady state behavior with respect to relative position, manner of oscillation, and current. One tree was then moved out of this preferred location, reducing the total current through the system. Because this perturbation reduces the current, it can be considered a functional impairment. Following the perturbation, the system relaxed back to steady-state dynamics. During the relaxation phase, the current through the system increased as the trees moved. Further, cross-recurrence quantification analysis (CRQA) of the tree motion showed that the degree of activity of each tree was coordinated over the relaxation phase [40]. Together, the results suggested that the trees were functionally interdependent, coordinating their behaviors to increase the current.

### 1.4. The Present Study

Building on previous work, we aim here to test whether a pair of E-SOFI trees will exhibit reciprocal compensation akin to that observed in biological coordinative structures. We show that the E-SOFI system exhibits each of the three properties of a coordinative structure described above [11,13] and that coupled trees can compensate for perturbations. To do so, we explore the dynamics of a two-tree E-SOFI system in which the trees are coupled through a shared distribution of charges on the oil surface. We use two separate grounding electrodes to constrain the relative position of the trees and to measure each tree’s individual contributions to the system’s current (Figure 2).

Previous work has demonstrated that the activity of a single tree is driven by the interplay of the distribution of charges on the oil and the depletion of charges by the tree [31]. The tree and the charge-distribution are mutually constraining: the tree conducts charges and changes the distribution, which, in turn, changes the forces on the tree driving its motion. Given the relationship between a single tree and the charge distribution, when two trees are present, they should be coupled through the shared charge distribution. Each tree’s activity is driven by the distribution of charges, which, in turn, is shaped by the activity of both trees. We can test this account of coupling by manipulating the distance between the two trees; trees that are further apart will be more weakly coupled, due to the spatial dependence of electrical forces. Thus, we predict bi-directional coupling between trees, which would satisfy property (1) for a coordinative structure.

Previous work has also shown that the dynamics of a single tree can change as a function of context [19,31]. Here, we predict that when one of the two trees has its movement restricted by a magnetic field, the other tree will compensate with a change in its motion, thus satisfying property (2) of flexibility. We observed that this flexibility is most apparent when the trees are strongly coupled.

Given that the trees are rudimentarily end-directed to maximize the REP [21], activities that contribute to this end may be considered functional in that the behaviors emerge to maximize the REP. Thus, when one tree is locked down, the compensation of the other tree should not only be evident in the tree motion, it should also be reflected in the contribution to the system’s REP. This would fulfill property (3) of a coordinative structure.

In summary, in the current study we created an analog experimental paradigm for investigating reciprocal compensation within a non-living dissipative system composed of two self-organized structures. We impose a functional constraint on Tree 1 (top tree, Figure 2) by imposing a magnetic force that limits its motion and measure changes in the dynamics and functionality of Tree 2 (bottom tree, Figure 2). Each trial consists of two phases wherein Tree 1 is either freely oscillating (“Unlocked”) or magnetically constrained (“Locked”). If the two trees behave as a coordinative structure, we predict that, when Tree 1 is locked down, Tree 2 should exhibit: a) a change in its motion; and b) an increase in the current flowing through it. To quantify the motion of the trees, we measure each tree’s mean displacement from the source electrode and mean oscillation amplitude. The restriction of movement of Tree 1 should decrease the current flowing through it by restricting its access to the charge-distribution. Crucially, we predict that Tree 2 will compensate for the restriction on Tree 1 by changing its motion and thus have increased current flowing through it.

## 2. Materials and Methods

### 2.1. Materials and Procedure

Two trees of 5 beads each, set by hand, were placed on grounding electrodes in a shallow bath of oil (80 mL) inside a square dish (6″ × 6″). Tree 1 (top tree, Figure 2) was composed of 5 beads, 4 aluminum beads, and 1 chrome bead at the tip. Tree 2 (lower tree, Figure 2) was composed of 5 aluminum beads. The chrome bead was sensitive to magnetic fields, while the aluminum beads were not. All beads were 4 mm in diameter. The applied voltage was 26 kV, and the total current through both trees was in the range of 2–3 μA. The resulting ohmic heating had negligible impact on the temperature of the oil.

A magnet was positioned below the dish, initially at a distance removed such that its force was too weak to affect the dynamics of the tree. The magnet could be raised closer to the dish so that the chrome-tipped Tree 1 would be pulled towards it, constraining its motion. When the magnet was raised, it attracted the chrome-bead of Tree 1 to the extreme of its oscillatory trajectory such that it was maximally displaced from both the source electrode and Tree 2 (Figure 2B). While the magnet was raised, Tree 1 remained oriented away from the source, and its motion was largely restricted. Locking Tree 1 in this manner is intended to restrict its ability to draw current from charge-rich regions of the dish, which serves as a *functional* perturbation, given the intrinsic end to maximize the current (and thus REP). This period of magnetic influence on Tree 1 constitutes the “Locked Phase” of trials, while the period with the magnet in its lowered position constitutes the “Unlocked Phase”.

Trials consisted of three periods: a 10-min warm-up (to ensure steady-state dynamics), 10 min of unconstrained motion (the *Unlocked Phase*), and a 10-min perturbation period during which Tree 1 was magnetically locked and displaced from the source electrode (the *Locked Phase*). The current conducted by each tree was measured by a resistor probe on the grounding wire. Position data of the tip-beads of each tree were collected via deep-learning video processing [41].

The Locked and Unlocked phases were crossed with three conditions of varying degrees of coupling. Coupling was manipulated by varying the distance between trees, thus varying the degree to which the trees shared a pool of charges on the oil surface. Greater distance meant that the trees occupied more independent regions of the charge distribution, and thus the influence of one tree on the others’ local charge-distribution was smaller. Distance between trees was varied, while the tip of each tree’s distance from the source electrode was maintained by moving the grounding electrodes around a circle with a radius of 5 cm centered on the source electrode. In the *high-coupling* condition, grounds were separated by approximately 2 cm. In the *medium-coupling* condition, the grounds were at a nearly 90-degree angle separated by approximately 5.88 cm. In the *low-coupling* condition, the grounds were directly opposite each other (180 degrees) separated by approximately 10 cm (Figure 3).

### 2.2. Data Processing

To quantify the tree dynamics, we measured the displacement of tip-beads from the source, and the amplitude of oscillations with position data collected from video recordings. To quantify the functional aspect of behavior, we measured the current conducted by each tree. All means were taken over a 400-s subsection of each trial phase, immediately before and after the introduction of the magnet for the Unlocked and Locked phases, respectively. Current values are presented as z-scores (normalized at the trial level) to eliminate variation in baseline current due to properties of the air and oil.

For the displacement from the source, *x,y* position data of the tip-bead of each tree were converted into a scalar Euclidean distance from a point in the dish that was minimally displaced from the source. To measure the oscillation amplitude, we converted *x,y* position data into a scalar timeseries consisting of the distance from the left extreme of the trees’ cycles (i.e., the furthest in the clockwise direction). These timeseries were converted to cycle phase in radians via a forward-backward Butterworth filter and a Hilbert transform to produce the analytic signal [42]. Oscillation amplitude was computed as the absolute value of the analytic signal.

## 3. Results

### 3.1. High-Coupling Condition

Figure 4 displays a sample subsection of timeseries of the current conducted by Tree 1 and Tree 2. This subsection is from a 200-s portion of the Unlocked Phase of one trial in the high-coupling condition. The current oscillates due to the oscillations of the trees (see [31] for details of the oscillatory dynamics). The REP is directly proportional to the current and is similarly time-varying due to the oscillatory dynamics. Figure 4 illustrates the time-varying characteristics of the current, though subsequent analyses average over these oscillatory cycles within trial phase (i.e., with Unlocked and Locked phases).

In the high-coupling condition, Tree 1 was on average closer to the source during the Unlocked Phase (*M* = 35.802 mm, *SD* = 1.59 mm) than the Locked Phase (*M* = 42.357 mm, *SD* = 0.365 mm), (*t*(3) = −10.604, *p* < 0.05), (Figure 5A). Additionally, the normalized current through Tree 1 was higher in the Unlocked Phase (*M* = 0.376, *SD* = 0.0523) than the Locked Phase (*M* = −0.379, *SD* = 0.055), (*t*(3) = 14.121, *p* <0.05), (Figure 5D). One-tailed, paired samples t-tests are used to test all between-phase (Locked vs. Unlocked) differences. Throughout, the effects on Tree 1 are largely a manipulation check; locking the tree away from the source with a magnetic field should produce consistent effects. Oscillation amplitude is not reported for Tree 1 because the oscillations are largely damped out during the Locked Phase.

Tree 2 was on average further displaced from the source during the Unlocked Phase (*M* = 36.064 mm, *SD* = 1.064 mm) than the Locked Phase (*M* = 35.504 mm, *SD* = 1.122 mm), (*t*(3) = 4.0425, *p* < 0.05) (Figure 6A). Similarly, a t-test of Tree 2’s oscillation amplitudes between trial phases in the high-coupling condition revealed a significant difference between the Unlocked (*M* =7.723 mm, *SD* = 1.699 mm) and Locked (*M* = 8.201 mm, *SD* = 1.951 mm) phases, (*t*(3) = −2.948, *p* < 0.05), with an average increase in amplitude of 0.478 mm (Figure 6D). Finally, the normalized current through Tree 2 was lower in the Unlocked Phase (*M* = −0.276, *SD* = 0.0442) than the Locked Phase (*M* = 0.277, *SD* = 0.045), (*t*(3) = −12.359, *p* < 0.05) (Figure 6G). In accord with MEP, locking down Tree 1 reduced the current flowing through it and caused the behavior of Tree 2 to change, in both its mean displacement and oscillation amplitude, resulting in an increase in the current flowing through Tree 2. In other words, Tree 2 appears to draw more current in order to *compensate* for the loss of current in Tree 1.

### 3.2. Medium-Coupling Condition

In the medium-coupling condition, Tree 1 was on average closer to the source during the Unlocked Phase (*M* = 31.387 mm, *SD* = 1.981 mm) than during the Locked Phase (*M* = 33.587 mm, *SD* = 3.281 mm), (*t*(3) = −3.328, *p* < 0.05) (Figure 5B). The normalized current through Tree 1 was greater during the Unlocked Phase (*M* = 0.393, *SD* = 0.201) than during the Locked Phase (*M* = −0.393, *SD* = 0.201), (*t*(3) = 3.921, *p* < 0.05) (Figure 5E).

Tree 2 was on average closer to the source during the Unlocked Phase (*M* = 31.368 mm, *SD* = 1.815 mm) than during the Locked Phase (*M* = 31.806 mm, *SD* = 1.843), (*t*(3) = −6.925, *p* < 0.05) (Figure 6B). The oscillation amplitude for Tree 2 was significantly higher in the Locked Phase (*M* = 12.212 mm, *SD* = 1.791 mm) than in the Unlocked Phase (*M* = 11.637 mm, *SD* = 1.500 mm), (*t*(3) = −2.88, *p* < 0.1) (Figure 6E), with an average increase of 0.576 mm. Thus, contrary to our expectations, the average position of Tree 2 was actually *further* from the source during the Locked Phase. However, the average amplitude was greater. The normalized current through Tree 2 was marginally lower during the Unlocked Phase (*M* = −0.149, *SD* = 0.127) than during the Locked Phase (*M* = 0.150, *SD* = 0.127), (*t*(3) = −2.351, *p* > 0.05) (Figure 6H). Thus, as predicted, the current through Tree 2 increased when Tree 1 was locked down.

### 3.3. Low-Coupling Condition

In the low-coupling condition, Tree 1 was on average closer to the source during the Unlocked Phase (*M* = 32.236 mm, *SD* = 0.477) than during the Locked Phase (*M* = 35.881 mm, *SD* = 0.691), (*t*(3) = −6.421, *p* < 0.05) (Figure 5C). The normalized current through Tree 1 was greater during the Unlocked Phase (*M* = 0.384, *SD* = 0.128) than during the Locked Phase (*M* = −0.384, *SD* = 0.128), (*t*(3) = 5.999, *p* < 0.05) (Figure 5F).

There was no significant difference in Tree 2’s displacement from the source between Unlocked (*M* = 35.483 mm, *SD* = 0.675) and Locked (*M* = 35.661 mm, *SD* = 0.506 mm) phases, (*t*(3) = −0.802 *p* > 0.05) (Figure 5C). Likewise, Tree 2’s oscillation amplitudes between trial phases revealed no significant difference between the Unlocked Phase (*M* = 14.188 mm, *SD* = 0.406 mm) and the Locked Phase (*M* = 14.273 mm, *SD* = 0.341 mm), (*t*(3) = −1.366, *p* > 0.05), with an average increase in amplitude of 0.0842 mm (Figure 6F). Thus, in the low-coupling condition there were no significant differences in either the average position or movement of Tree 2 due to the status of Tree 1 (i.e., Locked vs. Unlocked.) Finally, there was no significant difference in Tree 2’s normalized current during Unlocked (*M* = 0.0259, *SD* = 0.075) and Locked (*M* = −0.0259, *SD* = 0.075) phases, (*t*(3) = 0.687, *p* > 0.05) (Figure 6I). The current flowing through Tree 2 did not increase when Tree 1 was locked down.

### 3.4. Coupling-Condition Effects on Inter-Phase Changes

We performed trend analyses to determine the effect of coupling level on the magnitude of inter-phase (from Unlocked to Locked) changes in displacement from the source (Figure 7A), oscillation amplitude (Figure 7C), and current (Figure 7B), for both Trees 1 and 2. These inter-phase changes are a coarse measure of each Tree’s response to the perturbation. There was a significant negative linear trend (*F*(1, 9) = 11.13; *p* <0.05) of Tree 1’s inter-phase displacement change across coupling levels. There was a significant positive linear trend (*F*(1, 9) = 11.30; *p* < 0.05) of Tree 2’s inter-phase displacement change across coupling levels. There was no linear trend (*F*(1, 9) = 0.004; *p* > 0.05) of Tree 1’s inter-phase current change across coupling levels. There was a significant negative linear trend (F(1, 9) = 22.93; *p* < 0.05) of Tree 2’s inter-phase current change across coupling levels. Thus, the manipulation of the coupling between trees did not affect the functional consequences of the perturbation to Tree 1 since the inter-phase changes in current were not different across coupling levels. The variation of coupling did drive changes in Tree 2’s response in accordance with expectations of the degree of coupling—weaker coupling resulted in a weaker response.

### 3.5. Simulations of the E-SOFI Dynamics

Self-organization, as understood by contemporary non-equilibrium thermodynamics, is driven by the nonlinear interactions between thermodynamic forces and flows [20,26]. In the E-SOFI, the driving force is the variation in electric potential across the system (i.e., the distribution of charges on the oil surface). The corresponding flow is the current of charges through the oil and trees to the ground. We used a computational model of the system representing these electrical forces and flows and simulated analogous perturbation experiments to demonstrate reciprocal compensation. The one-dimensional model, built and simulated in Matlab, consists of coupled differential equations representing a distribution of charges in a one-dimensional space and the resulting forces on the tip-beads of individual trees moving in that space. The model, originally describing the dynamics of a single tree, was extended to include two trees. Details of the single-tree model are presented in [31].

The model runs in a one-dimensional space *x(i)* consisting of *i = 1* to *i = n* discrete locations (*n* typically set to 2000). The space is defined to have the midpoint at *x* = 0, with equal extent on either side (i.e., *x* = {−10, 10}) The model consists of three coupled differential equations, one governing the distribution of charges *y(i)* over each location *x(i)* (Equation (2)), the other two representing the electrical forces on each bead generated by the charge distribution (Equations (3) and (4)).
(2)$$\dot{y}\left(i\right)=-\frac{c_{1}*y\left(i\right)}{{\left(x_{i}-x_{b1}\right)}^{2}+c_{2}}-\frac{c_{1}*y\left(i\right)}{{\left(x_{i}-x_{b2}\right)}^{2}+c_{2}}+\sigma \left(Cmax_{i}-y\left(i\right)\right)$$
(3)$${\ddot{x}}_{b1}=-\beta \cdot {\dot{x}}_{b1}+\left(p\cdot \nabla \right)E+q_{1}E\left(x_{b1}\right)+M\left(x\right)+q_{2}{\vec{r}}_{12}+f_{con1}$$
(4)$${\ddot{x}}_{b2}=-\beta \cdot {\dot{x}}_{b2}+\left(p\cdot \nabla \right)E+q_{1}E\left(x_{b2}\right)+q_{1}{\vec{r}}_{21}+f_{con2}$$

In Equation (2), *y(i)* is the amount of charge at location $x_{i}$, *c*_1_ is a constant between 0 and 1 that represents the conductivity of the bead, *x_b_*_1_
*and x_b_*_2_ are the locations of Beads 1 and 2, and *c*_2_ is a constant that prevents the denominator from going to zero. *Cmax_i_* sets the maximum saturation capacity for charges at each location *x_i_*, and *σ* is a constant that takes values between 0 and 1 and scales the saturation rate. The first two terms represent the depletion of charges by the beads, and the third represents the supply of charges from the source electrode.

Equations (3) and (4) represent the forces on each bead due to viscous damping, the force on the dipole due to an inhomogeneous electric field, Coulomb forces from the charge distribution, a Coulomb force between the charged beads, a magnetic force on Bead 1, and a constraint to restrict the beads’ motion. *β* is a coefficient of viscous damping due to the oil, *p* represents the electric field-induced dipole moment of the bead, *q*_1_ and *q*_2_ are the charges on beads 1 and 2, *E(x_bi_)* is the electric field at the bead’s location due to all charges in the system, *M(x_b_*_1_*)* is the magnetic force, $q_{2}{\vec{r}}_{12}$ is the Coulomb force between the beads, and *f_con_* is a force representing a physical constraint (explained below).

The field induces a dipole moment on each bead. The dipole is assumed to be aligned in the same direction as the field (which is assumed to the along the x-axis). Since *E* and *p* are along the x-axis, the force, (p·∇)E = p_x_(dE_x_/dx), is approximated from the electric field vector on either side of the dipole. Three terms are calculated representing the field at the bead, $E(x_{b})$, and the field to the left and right of the bead, $E(x_{b-1})$ and $E(x_{b+1}),$ respectively, each calculated as the sum of Coulomb forces from all charges in the charge-distribution. The gradients to the left and right of the bead are then calculated according to: $Lgrad=E(x_{b})-E(x_{b-1})\;Rgrad=E(x_{b+1})-E(x_{b})$. These two terms are averaged and multiplied by the dipole moment *p* to give the force. ${\vec{r}}_{12}$ is signed (positive or negative) according to the relative position of the beads. e.g., if Bead 2 is to the right of Bead 1 (i.e., $x_{b1}<x_{b2}$) then the Coloumb force on Bead 2 is positive, while the force on Bead 1 is negative. This simulates the beads repelling each other due to their shared negative charge.

The charge distribution is modeled as having a peak in the middle of the space at *x* = 0, with charges building up at a greater rate nearer this peak. This is analogous to the geometry of the E-SOFI with respect to the source electrode: the electrode is centered in the dish and charges accumulate on the oil to a greater degree nearer the source. Bead dynamics are thus presented as displacement from the source by taking the position values as displacement from *x* = 0. Perturbations are done with respect to the source such that Bead 1 is pulled away from this peak analogous to the experiments conducted with the E-SOFI.

In the E-SOFI, the grounding electrodes have insulating constraints that restrict the base bead of the trees, reducing the tree activity to a sweeping arc pivoting on the base bead. We impose an analogous constraint in the form of a position-dependent spring-force with high stiffness *k* conditional on the bead being within a specified range of the prescribed constraint position *x_c_* (Equation (5)). This functions like a wall, only generating force when the bead reaches the prescribed positions *x_c_* on either side of the bead.
(5)$$f_{con}=-k*\left(x_{b}-x_{c}\right)$$

By varying the position of these constraints, we can restrict the beads to subspaces of varying distance analogous to the coupling conditions in the E-SOFI. A given bead draws charges from all over the distribution and will draw more charges from nearer regions. The rate of conduction depends on the inverse square of the distance between the bead and the location of charges (Equation (2)). Bead 1 then will draw more charges from the region of the Bead 2 if the two are near each other (and vice versa) than if they are far apart. While these manipulations do not reflect the full geometry of the real system, they do capture the hypothesized mechanism of coupling in the mutual influence exerted through the charge-distribution.

We include a force representative of the magnet on Bead 1 to enable simulations of the perturbation experiments performed with the E-SOFI. The magnetic force is represented as an inverse-square equation dependent on the distance between the bead $x_{b1}\;$and the prescribed magnet location $x_{mag}$. $f_{M}$ is an arbitrary constant scaling the magnitude of the force (strength of the magnetic field) that is used to turn the magnetic force on or off, and *c* is a constant to prevent the denominator from going to zero.
(6)$$M=\frac{f_{M}}{{\left(x_{b1}-x_{mag}\right)}^{2}+c\;}$$

Simulations consist of two phases, an ‘Unlocked Phase’ wherein both beads are freely oscillating, followed by a ‘Locked Phase’ wherein the magnetic force is turned on and Bead 1 is consequently constrained. From the simulations, we obtain time-series of each tree’s displacement from the mid-point of the space (peak of the charge distribution) and the current drawn by each tree. Similar to the experiments with the E-SOFI, we perform the perturbation under three coupling conditions—high, medium, and low—by varying the distance between the beads and consequently the degree to which they draw from similar regions of the charge distribution (see simulation parameters in Table 1).

### 3.6. Simulation Results

Figure 8 and Figure 9 show the mean displacement, amplitude, and current for Beads 1 and 2, respectively, between trial phases and across coupling conditions. We observe that Bead 1 is consistently perturbed, being displaced from the source and having reduced current during the locked phase (Figure 8). Tree 2, mirroring the E-SOFI results, demonstrates decreased displacement, increased amplitude, and increased current, during the locked phase of each simulated trial (Figure 9). All simulations were deterministic, and thus have no statistical variability.

We observe changes in the inter-phase differences of displacement, current, and Bead 2 oscillation amplitude like those observed in the E-SOFI. Bead 1’s inter-phase change in displacement increased slightly with decreasing coupling (Figure 10A), unlike the decrease observed in the E-SOFI. The magnitude of Bead 1’s inter-phase change in current increased slightly (Figure 10B). This increase of the magnitude of the change (values were increasingly negative) indicated that as coupling was reduced, the functional perturbation had *greater impact*.

Bead 2’s inter-phase change in displacement increased with decreasing coupling (Figure 10A) like the E-SOFI. Bead 2’s inter-phase change in current decreased with decreasing coupling (Figure 10B), like the results from the E-SOFI. Bead 2’s inter-phase amplitude change decreased with decreasing coupling (Figure 10C). The simulated amplitude and current data corroborate the E-SOFI data, supporting the interpretation that Bead 2 exhibited smaller functional changes when the beads were more weakly coupled.

## 4. Discussion

### 4.1. Interpreting the Results

In the present experiments and simulations, we investigated the coordinative properties of coupled dissipative structures, demonstrating that they exhibit reciprocal compensation. The E-SOFI exhibits three properties of biological coordinated behavior: (1) multiple couple constituents (the trees); (2) flexibility (context-dependent dynamics); and (3) intrinsic functionality (maximization of the REP). Both dissipative structures share an intrinsic aim to maximize the REP (i.e., current). When one tree is perturbed and is functionally impaired with respect to that aim, we observe compensatory activities in the other tree, changing its behavior in a way that increases the current it draws and consequently the REP. These results mirror the phenomenon of reciprocal compensation observed in biological instantiations of intra- and inter-personal coordination. Whereas coordination in biology may often be attributed to complex physiochemical processes, *the results here demonstrate that coordinated behavior can emerge in simple physical systems*. An intrinsic end-directedness—here to maximize the REP—supports sophisticated life-like behaviors without invoking complex biological mechanisms.

During the Locked Phase of the experiments (i.e., when Tree 1 was locked down), Tree 2 changed its behavior in two ways: by adjusting its distance from the source electrode and by increasing its oscillation amplitude. We observed slightly different changes in dynamics for the different coupling conditions, which were likely driven by the changes in geometry. In the high-coupling condition, Tree 2 reduced its displacement, while in the medium-coupling condition we observed an increase in the displacement. In the high-coupling condition, decreasing displacement meant moving into the charge-rich region between the two trees, thus collecting more charges. This decrease in displacement was accompanied by an increase in the oscillation amplitude. In the medium-coupling condition, the increase in displacement was accompanied by an increase in oscillation amplitude, increasing the range of the charge-distribution the tree accessed, but due to the orientation of grounds in the dish that meant increasing displacement from the source electrode. The oscillation amplitude then may be a more consistent measure of the tree’s dynamics, while the displacement from the source likely depends on the relative geometry of the trees. The low-coupling condition demonstrated no significant effects in either the displacement from the source or the mean oscillation amplitudes, in line with the expectation that no behavioral adjustment should occur in the low-coupling condition.

The degree to which Tree 2 compensated for Tree 1’s reduced functionality (as indexed by a reduction in the current flowing through Tree 1) depends on the degree to which the trees are coupled. The compensatory effect—that is, the increase in Tree 2’s current between Unlocked and Locked phases—was largest in the high-coupling condition, smaller in the medium-coupling condition, and not present in the low-coupling condition. There was no effect of coupling on the change in current for Tree 1, meaning that the functional consequences of the perturbation were consistent over coupling conditions. Interestingly, the fact that the low-coupling condition appears to essentially decouple the trees motivates that the trees are fundamentally distinct entities that can become coordinated when they share a field of constraints. This is not unlike the way that inter-personal coordinative structures can emerge and dissolve as our behaviors become entangled by shared environmental or social constraints.

Simulations of the system reveal a pattern of effects similar to that observed in the physical E-SOFI. At the behavioral level, Bead 2 demonstrated a decrease of displacement from the source (i.e., Bead 2 moved closer to the source) and an increase in oscillation amplitude when Bead 1 was locked down. The decrease in displacement of Bead 2 during the locked phase was greatest in the high-coupling condition, consistent with results from the physical system. In the medium- and low-coupling conditions, we observed less change in displacement but still a shift towards the source. In the E-SOFI, recall that Tree 2’s displacement increased (i.e., Tree 2 moved further from the source) in the medium-coupling condition. We suspect this difference between the model and the physical system is driven by the differences in geometry between the real and simulated systems, since the model can only accommodate one-dimensional changes in the distance between beads. The amplitude effects are, however, consistent between the E-SOFI and the simulated data. Oscillation amplitude may be the more consistent behavioral variable because it determines how much of the charge-distribution the tree accesses: increasing oscillation amplitude likely drives Tree 2’s increases in current observed during the Locked phase.

These behavioral changes corresponded with an increase in the current conducted by Bead 2. The degree of coupling was manipulated by constraining the beads to increasingly distant regions of the charge distribution. The magnitude of the compensatory effects in Bead 2 decreased with decreasing coupling, mirroring the results from the E-SOFI. The functional consequences of the perturbation increased with decreasing coupling (i.e., the magnitude of the decrease in Bead 1’s current), but the compensation by Bead 2 decreased, suggesting that the compensation depends on the coupling, not the magnitude of the functional impairment on Bead 1. While the model lacks many of the complexities of the E-SOFI, it captures the essential properties of the coupling between trees as a mutual influence exerted through a shared distribution of charges.

### 4.2. Entailments of a Thermodynamic Account

Self-organization has been hypothesized to be crucial to biological action by virtue of organisms being dissipative structures [20,24,29,30]. We build on this framework by demonstrating that dissipative structures can, in fact, behave like coordinative structures. Beyond supporting this hypothesis, these results highlight the physics that likely support coordinative structures. The tools of thermodynamics are attractive for their generality, as the quantities studied—energy, entropy, forces, and flows—have relationships that hold across a variety of processes including thermal, mechanical, electrical, and chemical systems [26] (Kondepudi and Prigogine, 1998). The many processes within an organism that enable behavior then can be described and studied in terms of thermodynamic quantities.

In the context of coordination, we identify three properties of thermodynamic systems that are sufficient to instantiate reciprocal compensation. First, the elements or constituents must be in a shared field of constraints. Second, the elements must be sensitive to this field such that changes in the field change the state of the elements. Third, the activity of each element must alter the shared field. We have elaborated on these properties elsewhere [43] in terms of both electrical and chemical forces and flows and review the coordinative phenomena that result from these physical processes.

In the E-SOFI, the elements are the trees and the shared field is the electrical field. The trees maintain themselves at ground; maintaining low electrical potential relative to the charge-rich oil creates the sensitivity to the electrical field, which in turn supports the continued existence of the trees. The charges in the field exert Coulomb forces on the grounded beads, whose magnitudes depend on the density and distribution of charges. Thus, if symmetry of the electrical field is broken (as it is when the trees conduct charges to ground), a corresponding asymmetry in the electrical forces drives motion of the tree. Lastly, each tree conducts charge to ground, thereby altering the shared field. These three properties have clear analogues in human interpersonal coordination as mediated by vision [44,45,46,47]. Individuals share an optic field (with physical components and social meaning) that constrains and informs behavior; individuals are sensitive to the structure of light by virtue of their organization; and their activity deforms the optic array, scattering light differently as they move.

### 4.3. Implications for Biological Coordination

This framework of self-organization has significant consequences for how we expect biology to behave. One immediate consequence of the present work is the possibility that aspects of coordination, for example, error compensation in active control, may arise by virtue of the physical organization of the system, rather than explicit control of physiological degrees of freedom. In the E-SOFI, error-compensation arises from the cross coupling of electrical flows through a shared electrical force—mutual modulation of the shared charge distribution necessitates functional interdependence of the trees. Another group [48] used a physical model—a hypothetical rusty bucket with a flow of water—to derive a description of error-compensation in the nervous system. Similar error-compensation will occur during intrapersonal biological coordination, for example, in the case of maintaining one’s grip on an object between the thumb and index finger. Perturbations to the force production of one finger will lead to compensatory changes in the other [49]. Similar phenomena have also been observed at the interpersonal level [50]. The biological instantiation of error compensation is, of course, more complex and intricate in detail than that of the E-SOFI. Present research on dissipative structures offers a potential pathway for scaling up the complexity of artificial life systems to approach more biologically plausible models.

## 5. Conclusions

Herein, we revisited a theoretical proposal that biological coordinative structures are a type of dissipative structure [28] and provided novel empirical evidence that non-living dissipative structures exhibit core properties and dynamics of biological coordinative structures. The three main properties of a coordinative structure are observed in the E-SOFI, and the system exhibits compensatory coordination such as biological dissipative structures. The coordination is tied to the system’s intrinsic aims to maximize the REP; the tree’s behaviors are coordinated by virtue of a shared “goal” to maximize the REP, much like the joint action of organisms with shared intentions. Crucially, the finding that these properties are generic to non-living dissipative structures invites consideration of the physical basis of coordination in living dissipative structures broadly. These physical and dynamical processes of coordination are essential participants in the perception, action, and cognition underwriting the control of action, including social action.

## Figures and Tables

**Figure 1 entropy-23-00614-f001:**
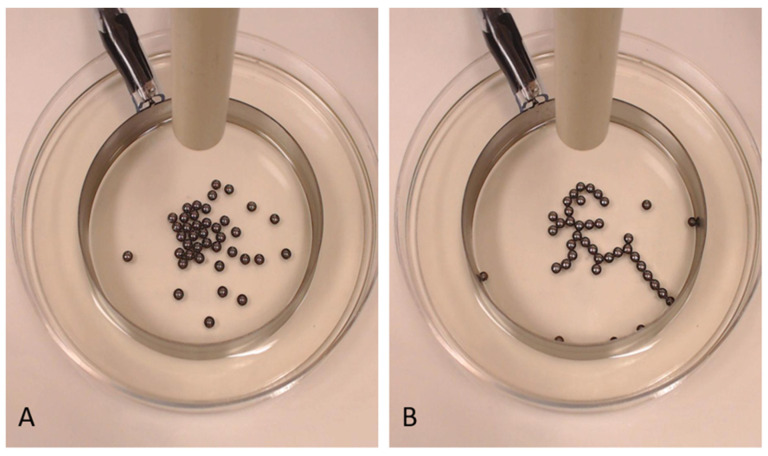
The E-SOFI, (**A**) pre- and (**B**) post-tree formation. Charges are supplied from an electrode above the dish. The structure in (**B**) only lasts if the system has a flow of charges available.

**Figure 2 entropy-23-00614-f002:**
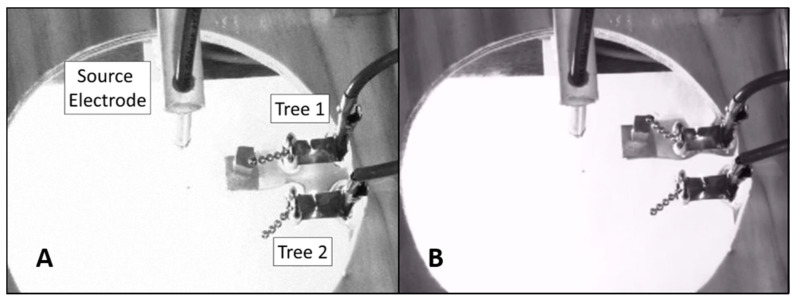
(**A**) The two-tree E-SOFI. Separate grounding electrodes are fixed to the bottom of the glass dish. The source electrode is above the dish. Wires connect the electrodes to the ground on the power supply. A magnet on a moveable arm is visible below the dish. (**B**) The magnet is raised near the dish, locking Tree 1 to the outside extreme of its oscillatory cycle.

**Figure 3 entropy-23-00614-f003:**
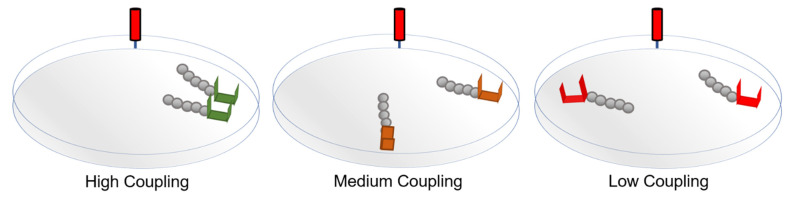
Schematics of the dish setups to for each coupling condition. The distance between the front of the grounding brackets and the source was maintained across coupling levels.

**Figure 4 entropy-23-00614-f004:**
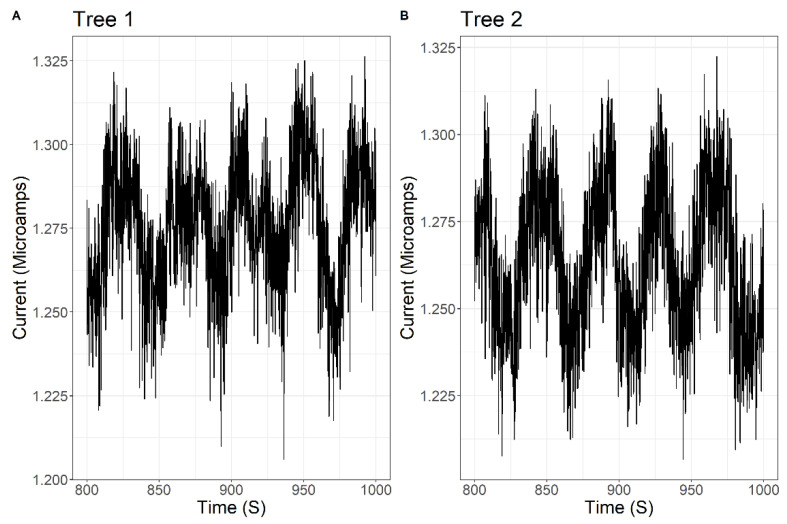
Sample time-series of the currents for both (**A**) Tree 1 and (**B**) Tree 2. Data are taken from a 200-s subsection of the Unlocked Phase in the high-coupling condition. The current oscillates due to oscillations of the trees. The REP is proportional to the current and similarly oscillates with the current.

**Figure 5 entropy-23-00614-f005:**
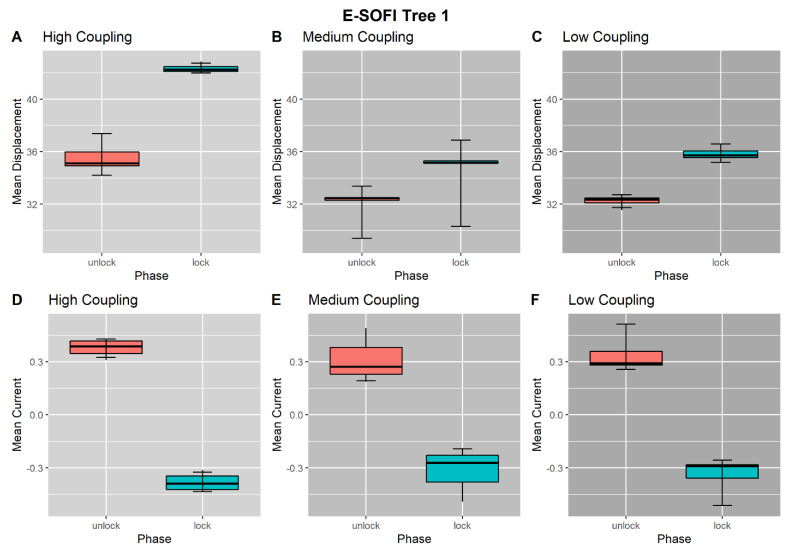
(**A**–**C**) Tree 1’s average displacement from the source electrode, within each Unlocked and Locked phase, across coupling levels. Tree 1 consistently is further displaced from the electrode in the Locked Phase due to the magnetic constraint. This effect is present across coupling levels. (**D**–**F**) Tree 1’s average current conducted within each Unlocked and Locked phase, across coupling levels. The current conducted by Tree 1 decreases in the Locked Phase, due to the magnetic constraint. This effect is present across coupling levels.

**Figure 6 entropy-23-00614-f006:**
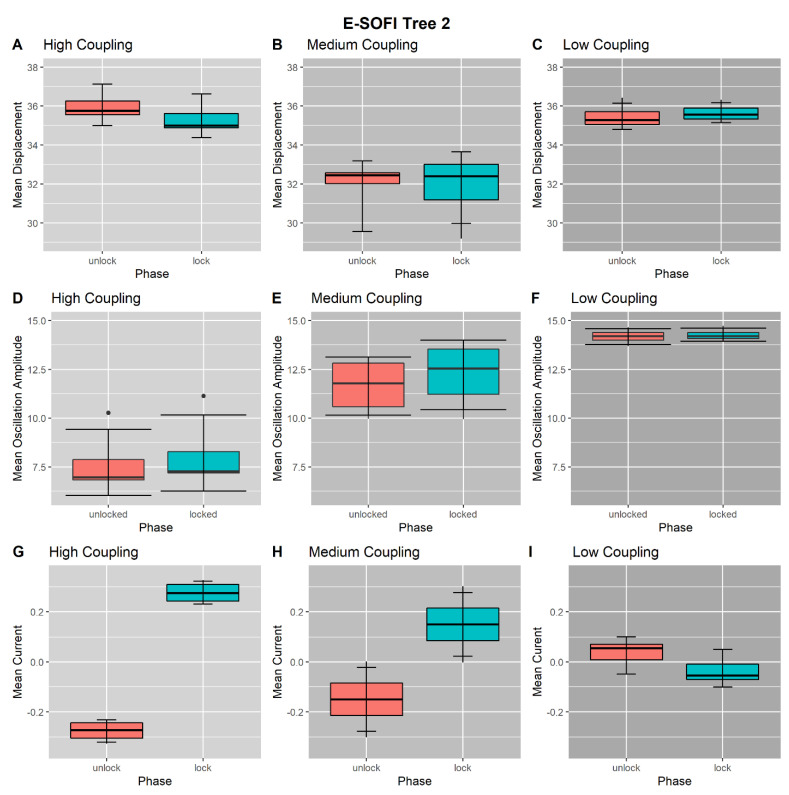
(**A**–**C**) Tree 2’s average displacement from the source electrode within Unlocked and Locked Phases, across coupling levels. In the high coupling condition, Tree 2 is less displaced from the source during the Locked Phase. In the medium coupling condition Tree 2 is more displaced from the source during the Locked Phase. In the low coupling condition, there is no difference in displacement between phases. (**D**–**F**) Tree 2’s average oscillation amplitude within Unlocked and Locked phases, across coupling levels. In the high and medium coupling conditions, Tree 2’s oscillation amplitude increases during the Locked Phase, while there is no difference in the low coupling condition. (**G**–**I**) Tree 2’s average current within Unlocked and Locked phases, across coupling levels. In the high and medium coupling conditions Tree 2’s current increases during the Locked phase, while in the low coupling condition Tree 2’s does not change. Together, these results suggest that Tree 2 has increased current during the Locked Phase, compensating for the reduction in current from Tree 1, facilitated by an increase in oscillation amplitude.

**Figure 7 entropy-23-00614-f007:**
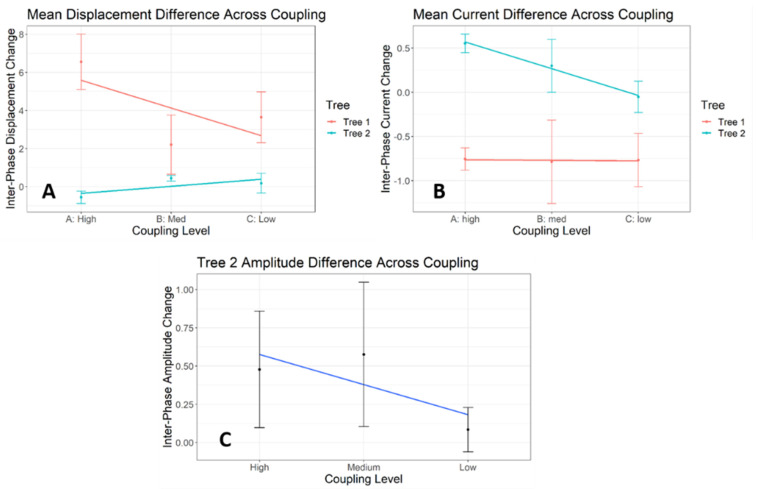
The coupling level is intended to modulate the magnitude of the compensatory response of Tree 2, with an expected decreasing response as coupling decreases. (**A**) Mean inter-phase difference (Locked–Unlocked) in displacement across coupling levels. (**B**) Mean inter-phase differences (Locked–Unlocked) in current across coupling levels. Tree 1’s change in current is consistent across coupling levels, suggesting the perturbation has similar functional impacts across coupling levels. Tree 2 exhibits a clear decrease in the inter-phase current change, suggesting that it is less able to compensate when the trees are less strongly coupled. (**C**) Mean inter-phase differences (Locked–Unlocked) in Tree 2’s oscillation amplitude across coupling levels. The inter-phase change in oscillation amplitude decreases with decreasing coupling, suggesting that Tree 2 changes its dynamics less when the trees are less strongly coupled.

**Figure 8 entropy-23-00614-f008:**
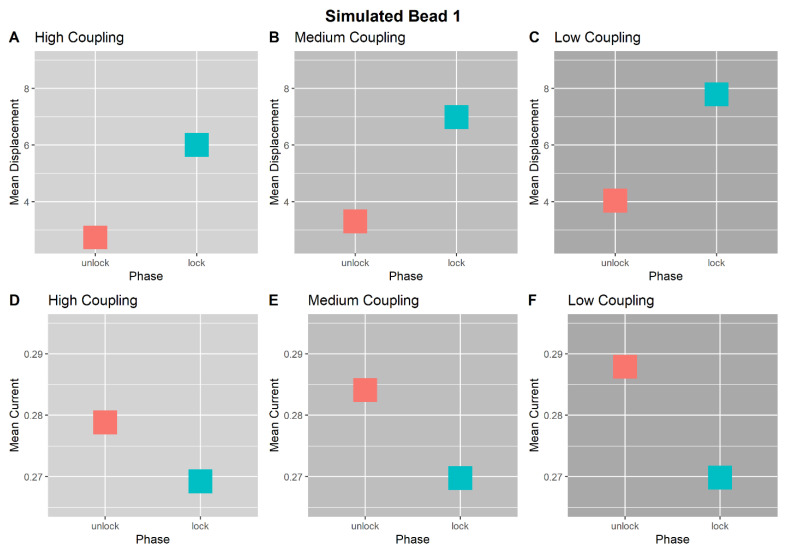
Results of computer simulation. Bead 1 simulated displacement and current between trial phases and across coupling conditions. Results neatly mirror those of the E-SOFI. Simulations were run deterministically, with only one trial per coupling level. The above plots include boxes for clarity, and do not represent variability in any measures. Simulations assume constant voltage and temperature, and thus the REP is proportional to the current. (**A**–**C**) Bead 1’s average displacement from the peak of the charge-distribution, within each Unlocked and Locked phase, across coupling levels. Bead 1 consistently is further displaced from the electrode in the Locked Phase due to the magnetic constraint. This effect is present across coupling levels. (**D**–**F**) Bead 1’s average current conducted within each Unlocked and Locked phase, across coupling levels. The current conducted by Bead 1 decreases in the Locked Phase, due to the magnetic constraint. This effect is present across coupling levels.

**Figure 9 entropy-23-00614-f009:**
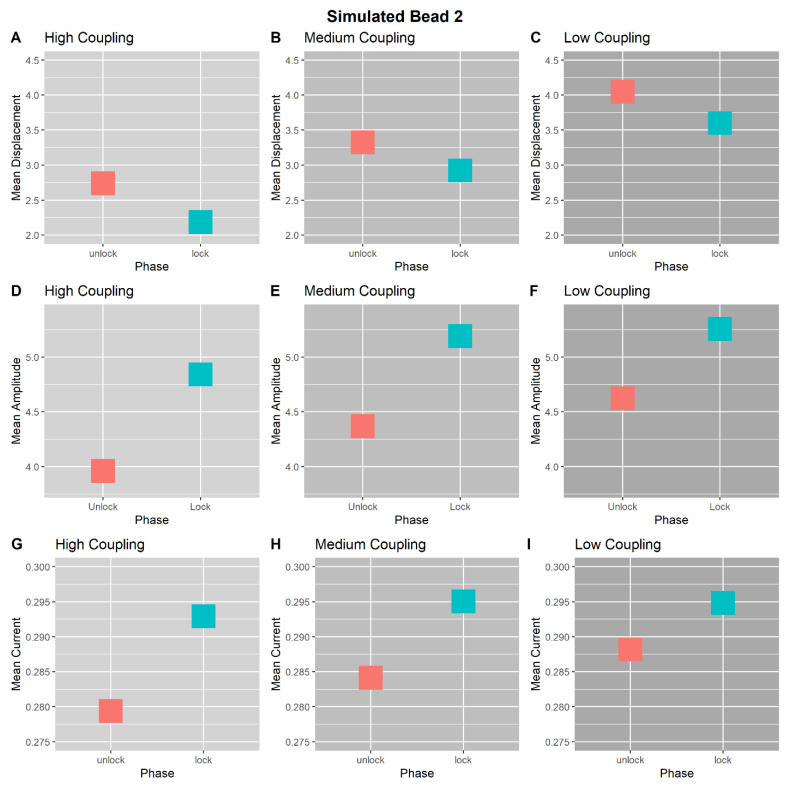
Results of computer simulation. (**A**–**C**) Bead 2’s average displacement from the peak of the charge-distribution within Unlocked and Locked Phases, across coupling levels. In all three coupling conditions, Bead 2 was less displaced from the peak in the Locked Phase than in the Unlocked Phase. (**D**–**F**) Bead 2’s average oscillation amplitude within Unlocked and Locked phases, across coupling levels. In all coupling conditions, Bead 2’s oscillation amplitude increased during the Locked Phase. (**G**–**I**) Bead 2’s average current within Unlocked and Locked phases, across coupling levels. In all coupling conditions, Bead 2’s current is greater in the Locked Phase. Together, these results suggest that Bead 2 had increased current during the Locked Phase, compensating for the reduction in current from Bead 1, facilitated by an increase in oscillation amplitude.

**Figure 10 entropy-23-00614-f010:**
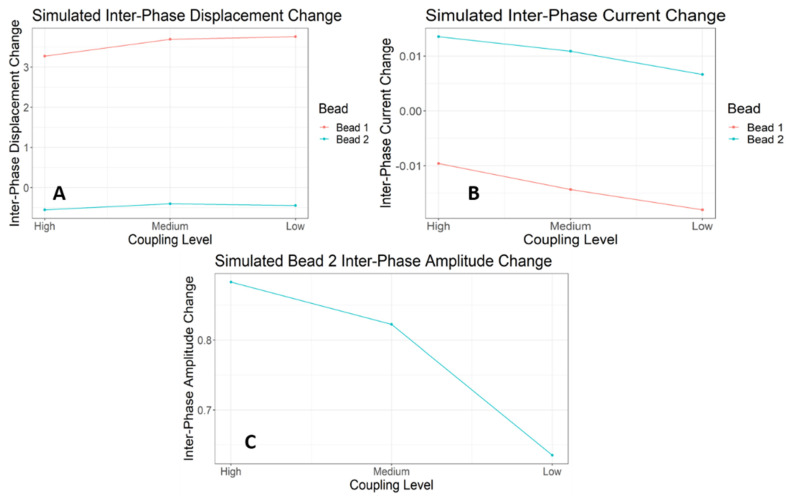
The coupling levels were intended to modulate the magnitude of the compensatory response of Bead 2, with an expected decreasing response as coupling decreased. (**A**) Mean inter-phase difference (Locked–Unlocked) in displacement across coupling levels. (**B**) Mean inter-phase differences (Locked–Unlocked) in current across coupling levels. Bead 1’s change in current becomes *increasingly negative*, meaning the perturbation had increased functional consequences with decreased coupling. Bead 2 exhibits a clear decrease in the inter-phase current change, suggesting that it is less able to compensate when the trees are less strongly coupled. This is despite the increased perturbation to Bead 1. (**C**) Mean inter-phase differences (Locked–Unlocked) in Bead 2’s oscillation amplitude across coupling levels. The inter-phase change in oscillation amplitude decreases with decreasing coupling, suggesting that Bead 2 changes its dynamics less when the trees are less strongly coupled.

**Table 1 entropy-23-00614-t001:** Simulation parameters.

s	High Coupling	Medium Coupling	Low Coupling
Bead 1 Constraints	*x_c_ =* (−3, −1)	*x_c_ =* (−4, −2)	*x_c_ =* (−5, −3)
Bead 2 Constraints	*x_c_ =* (1, 3)	*x_c_ =* (2, 4)	*x_c_ =* (3, 5)

## Data Availability

The data presented in this study are available through Open Science Framework, doi:10.17605/OSF.IO/7FMKS.

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
