# Peer review of "Functional Interdependence in Coupled Dissipative Structures: Physical Foundations of Biological Coordination"

_entropy, 2021, doi:10.3390/e23050614_

Round 1
Reviewer 1 Report
The manuscript under review gives some interesting insight to the mechanisms by which coordinated cooperative behaviours can emerge in complex systems. In particular, the authors focus on the role of coordinative structures, which rise during self-organized processes.
The idea to test this framework by means of non-living structures is particularly nice because helps find the very basic ingredients which may have a role in this kind of social (in broad sense) phenomena. The role of simple physical organization instead of more complex features present in living systems deserves indeed more attention.
The intrinsic property of a system to maximize the entropy production, so that a component intervenes if another one undergoes any problem, seems to be also at the root of such processes.
In general I appreciated this work and repute it worth of publication. Nevertheless, there are some minor issues that I suggest to fix before publication:
1) In the text references are indicated by numbers, but in the Bibliography the numbers do not appear, making hard to find each time the righ reference;
2) Some figures are not well formatted: Figure 3, Figure 4, Figure 5, Figure 9 and Figure 10;
3) Table 1, with the list of simulation parameters, is invisible;
4) No error bars in Figures 11-12-13?
5) It is not totally clear to me the meaning of Unlocked and Locked phases. What does mean exactly "magnetically locked and displaced from the source electrode"? What is the main difference of motion in the two phases (because I understand that the tree moves someway also in the Locked phase)?
6) Please clarify the formalism of equations (3) and (4) at page 11: the second and third equations, are they scalar or vectorial? Since variables x_i appear to be unidimensional, what is the meaning of ∇E? Is this the divergence of E? Or the modulus of the spatial derivative of an unidimensional elctric field? Same for M(x) in (3) (why here x is not x_b1?). Finally, the distances r⃗_12 and r⃗_21 are the absolute values or their sign (positive or negative) must be taken into consideration?
Author Response
Thank you for your thoughtful suggestions. Please see the attached document for our responses and modifications to the manuscript.

Reviewer 2 Report
De Bari and colleagues have submitted a manuscript to journal Entropy entitled “Functional interdependence and coupled dissipative structures: physical foundations of biologic coordination” for consideration of publication. They evaluate the function of a “simple self organizing system“ comprised of metal beads in shallow oil subjected to a high electrical voltage. The beads self-organize them selves into trees, which oscillate gently, dissipating the electrical gradient from an outer charged ring. This “system” is called the “electric self-organized foraging implementation” and have been labelled with the acronym E-SOFI.
The authors have studied a system of two “trees” of five beads each, and applied external limitation to their dynamics with a magnet to constrain motion, and varied the proximity of the two trees, which they have called “coupling“. The interpretation of the results is that coordinated behaviour can emerge in simple physical systems, as the trees demonstrated compensation, provided there was proximity between the two trees. The authors conclude that “this coordinated activity of the system appears to derive from the systems intrinsic end-directed behaviour to maximize the rate of entropy production.” The authors also report a Matlab model of “trees” and report the results of the simulations.
The authors are to be commended for building and studying a simple functional model subject to experimental manipulation of interacting dissipative structures comprised of metal beads that can coordinate their interaction while dissipating an electrical gradient.
The following comments are intended to help improve the accessibility and impact of their paper.
Major comments:
While the authors conclude that the activity of the system appears to derive from the systems attempts to maximize the rate of entropy production, there is no experimental evidence that this is indeed the case. Of the 13 figures in the manuscript, none evaluate the rate of entropy production, either statically or over time. The authors provide the calculation of entropy production in equation 1, thus can entropy production be graphically represented for the evolution of this system? This data would greatly support the conclusion of the study, which are currently, as far as I can tell, unsupported by data on entropy production within this paper. This needs to be clarified.
Perhaps related to this first point, it appears the paper is a mix of theoretical conjecture, as well as both simulated and experimental results. The inclusion of all makes it very hard to discern the focus, or understand what precisely was performed. For example, the abstract makes no mention of the physical (or simulation) model that is being experimentally evaluated. Second, there are numerous results from simulated experiments. It is unclear why so many results are presented regarding coupling. Finally, there is a significant degree of generalizing of the findings to other domains of science such as social science. The problem with mixing all together is that it takes away from the impact of any of the components of the paper. The authors should clarify in the abstract that this is a initial combination of an experimental evaluation of a physical system, combined with a simulation, combined with a theoretical evaluation of the results, or separate the paper into its distinct components.
The experimental model could be expanded upon. Why limit the system to five beads only? Have the authors done experiments with more beads that simulate true multi-branched trees? Figure 1 demonstrates a model with numerous beads. Why not perform a model of two large trees? Is there a physical limitation to larger models?
In addition, the theoretical discussion regarding maximum entropy production (MEP) could also be expanded upon. Additional references would enhance the discussion. For example, the link between the tree-like structures of beads is supported by references hypothesizing a link between self-organized fractal tree-like structures (in time and space) with maximal entropy production (see Seely A, Macklem P, Chaos, 2012, and Seely A, Entropy 2020). Is this relevant to this experiment? If the two trees are coupled, then they may be acting as one combined system, seeking to optimize their combined entropy production. In addition, the references to other MEP papers are scant, and could be broadened (e.g. see Martyushev, L. Life and Evolution in Terms of Maximum Entropy Production Principle. Preprints 2020, 2020050017 (doi: 10.20944/preprints202005.0017.v1) to name just one of many.
Minor comments:
I suggest there are too many figures. Suggest to cut down and provide supplementary material online.
Author Response
Thank you for your thoughtful suggestions. Please see the attached document for our response and changes to the manuscript.

Round 2
Reviewer 1 Report
The Authors have addressed all my points raised in my report, so now I fully endors the manuscript for publication.
Reviewer 2 Report
The authors have responded well to suggestions and questions. The revision of the abstract will help readers to understand the scope and extent of the paper. The additional mention of applications of MEPP adds to the paper.
The only question that was not been addressed by the authors is the relationship between maximum entropy production and the self-organized fractal treelike structures of the bead trees. However, the focus of this paper is more on the interaction and motion of the tree-like bead structures, rather than their fractal structure.